# The History and Mystery of Alveolar Epithelial Type II Cells: Focus on Their Physiologic and Pathologic Role in Lung

**DOI:** 10.3390/ijms22052566

**Published:** 2021-03-04

**Authors:** Barbara Ruaro, Francesco Salton, Luca Braga, Barbara Wade, Paola Confalonieri, Maria Concetta Volpe, Elisa Baratella, Serena Maiocchi, Marco Confalonieri

**Affiliations:** 1Pulmonology Department, University Hospital of Cattinara, 34128 Trieste, Italy; francesco.salton@gmail.com (F.S.); paola.confalonieri.24@gmail.com (P.C.); marco.confalonieri@asugi.sanita.fvg.it (M.C.); 2ICGEB, Area Science Park, Padriciano, 34128 Trieste, Italy; braga@icgeb.org; 3City of Health and Science of Turin, Department of Science of Public Health and Pediatrics, University of Torino, 34128 Trieste, Italy; barbarawade@hotmail.com; 4Life Sciences Department, University of Trieste, 34128 Trieste, Italy; maria.volpe@icgeb.org (M.C.V.); serena.maiocchi@hotmail.it (S.M.); 5Department of Radiology, Department of Medicine, Surgery and Health Science, University of Trieste, 34128 Trieste, Italy; elisa.baratella@gmail.com

**Keywords:** epithelial cells, alveolar type II cells, idiopathic pulmonary fibrosis, chronic obstructive pulmonary disease (COPD), lung cancer, acute respiratory distress syndrome (ARDS), COVID-19 disease

## Abstract

Alveolar type II (ATII) cells are a key structure of the distal lung epithelium, where they exert their innate immune response and serve as progenitors of alveolar type I (ATI) cells, contributing to alveolar epithelial repair and regeneration. In the healthy lung, ATII cells coordinate the host defense mechanisms, not only generating a restrictive alveolar epithelial barrier, but also orchestrating host defense mechanisms and secreting surfactant proteins, which are important in lung protection against pathogen exposure. Moreover, surfactant proteins help to maintain homeostasis in the distal lung and reduce surface tension at the pulmonary air–liquid interface, thereby preventing atelectasis and reducing the work of breathing. ATII cells may also contribute to the fibroproliferative reaction by secreting growth factors and proinflammatory molecules after damage. Indeed, various acute and chronic diseases are associated with intensive inflammation. These include oedema, acute respiratory distress syndrome, fibrosis and numerous interstitial lung diseases, and are characterized by hyperplastic ATII cells which are considered an essential part of the epithelialization process and, consequently, wound healing. The aim of this review is that of revising the physiologic and pathologic role ATII cells play in pulmonary diseases, as, despite what has been learnt in the last few decades of research, the origin, phenotypic regulation and crosstalk of these cells still remain, in part, a mystery.

## 1. Introduction

Back in 1954, C.C. Macklin had already begun to hypothesize on some of the most important functions of the pneumocyte type II or alveolar epithelial type II (ATII) cells [1]. He suggested that these cells were able to secrete a material with important properties, i.e., one that created a low surface tension, enhanced inhaled particle clearance, was bacteriostatic and promoted the prevention of transudation of interstitial fluid into the alveolus. Indeed, he even reported that these cells proliferated after lung injury by osmium tetroxide fumes [1]. In 1974, Kikkawa and Yoneda carried out studies on rodents and reported the first successful isolation of a surfactant-producing epithelial cell from their lungs [2]. Others followed in the wake, such as Mason and Williams, who, in 1977, developed the concept of the ATII cell as a defender of the alveolus, providing further “food for thought” [3]. 

Since then, further research on primary cultured ATII cells isolated from a variety of species has provided a large volume of data [4,5,6,7]. This has led to these cells being assigned important biosynthetic, secretory, metabolic, host defense and repair/regenerative functions, evidencing the pivotal role ATII cells play in the maintenance of alveolar homeostasis [3,8]. 

It is known that ATII cells, that have secretory, proliferative and innate immune functions, are small, cuboidal cells with the anatomic features of active metabolic epithelial cells containing a high density of mitochondria and having special apical microvilli [8,9,10,11]. They account for about 15% of total cells but only about 5% of the alveolar surface area in the healthy adult human lung [12]. These cells have lamellar inclusions, which are the intracellular storage form of surfactant. Each ATII cell in the adult human lung has a surface area of 250 µm^2^, meaning that the epithelium is most likely the initial defense to inhaled particles and gases. Alveolar fluid also has some important characteristics, i.e., a very low volume, about 35 mL per adult human lung, with an acidic pH of about 6.9 [9].

Moreover, ATII dysfunction or dropout is associated with the pathogenesis of various parenchymal lung diseases that include, amongst others, Hermansky–Pudlak syndrome-related pulmonary fibrosis, idiopathic pulmonary fibrosis and chronic obstructive pulmonary disease [8,13,14,15,16,17,18,19]. ATI cells, which cover 95% of the alveolus but make up only 8% of the total cells in the healthy adult human lung, are elongated, flat cells that mediate gas exchange. Although their fetal progenitor is still to be defined, they are present at birth [1,2,3,7]. The ATII cells in the alveoli have long been thought to function as progenitor cells, as when there is ATI cell injury, adjacent ATII cells are stimulated to multiply and transdifferentiate into ATI cells [20,21]. 

So as to better study the contribution ATII cells make in the pathogenesis of pulmonary diseases, the isolation of viable ATII cells is a key factor. The purification of ATII cells is by no means easy, as numerous hurdles must be overcome, such as cell viability, the impurity of isolated cells and the consequent outgrowth by contaminating lung fibroblasts. Moreover, both the isolation method and the culture conditions, e.g., plate coating and special mediums, influence the maintenance of the ATII cell phenotype in vitro, which is by far the most critical aspect for the setup of meaningful functional assays [13]. 

## 2. Scientific Advances in the Study of Lung Anatomy

### 2.1. New Roles of the Alveolar Epithelial Cells 

The alveolar epithelium is made up of alveolar type I (ATI) and ATII cell types (Figure 1 and Figure 2) integrated into the epithelium by broad intercellular junctions, with ATI at the lateral membranes with a tight junction at the apical edge, while ATI leaflet extends across the septum to form linings on both sides in niches where there is no cell body [8,16].

As the epithelial basal membrane is tightly opposed to the endothelium cell layer in the thinnest regions of the gas-exchange surface, the efficiency of gas-exchange between epithelial and endothelial cells is increased.

Indeed, there is an apparently homogeneous mesh of very thin capillary vessels on the other side of the epithelial cells, made up of a monolayer endothelium surrounding each alveolus like a mosaic, forming the vast respiratory surface, across which oxygen transfers to the blood.

Several authors provided insights as to the topological complexity of the alveolar epithelium by serial block face scanning electron microscopy (SBF-SEM), also showing how an extremely broad and thin gas exchange surface (130 ± 12 m^2^ alveolar surface, 115 ± 12 m^2^ capillary surface, 0.62 ± 0.04 ηm^2^ double monolayer tissue barrier thickness) is forced into a limited space, thanks to a fractal folding forming crumpled space-filling continuum of alveoli lined to very thin capillaries and open to airways [22,23,24].

Single-cell sequencing allowed for the recognition of two intermingled endothelial cell types, which, similarly to the epithelium, line the alveolar capillary endothelium with distinct morphologies and functions. The aerocyte or aCap is specialized in gas exchange and the trafficking of leukocytes, and the second cell type, termed gCap (general capillary), is specialized in regulating vasomotor tone and serves as a stem/progenitor cell in capillary repair and regeneration [25]. 

Aerocytes and ATI, such as gCAP and ATII, coordinate themselves not only during normal gas exchange but also during development and when an injury occurs. Aerocytes and ATI emerge together as cell differentiation begins. During adult life, ATII and gCap cells act in a coordinated way as bifunctional stem/progenitor cells that also serve a physiological function. Separating the progenitor function of ATII and gCap cells from ATI and aerocytes is a useful mechanism for the preservation of the gas-exchange surface.

Moreover, lineage tracing studies have shown the heterogeneity of ATII cells that display differential capacity of proliferation and differentiation in both homeostatic and regenerative states. 

There are several types of stromal cells in the interstitial region, including mesenchymal cells, pericytes, endothelial cells and immune cells [26,27,28]. Together with the extracellular matrix (ECM), these cells constitute the “putative” niche for ATII stem cells in a discrete but dynamic microenvironment, maintaining the stemness potential and promoting appropriate cell fate and migration decisions [1,4,17,29]. A recent extensive cell atlas of the human lung defined the gene expression profiles and anatomical locations of 58 cell types, including 41 out of 45 already known cell populations and 14 previously unknown ones [29]. Recently, the work of Lefrançais et al. showed that the lungs are a primary site of terminal platelet production, and this observation proposed for the first time the lung as an organ with haemopoietic potential [30].

The discovery of cell type specialization in both alveolar epithelium and endothelium transformed our understanding of the structure, function, regulation and maintenance of this specialized air–blood interface in health and disease.

### 2.2. New Technologies to Explore the Function of Alveolar Epithelial Cells

#### 2.2.1. Lung-on-Chip

Organs-on-Chips are micro-engineered cell-based systems developed to resemble the multicellular dynamics, cell–cell interactions, vascular perfusion and microenvironments of real organs. The lung-on-chip is a biomimetic micro-system able to reconstruct the functional alveolar epithelial–capillary interface [31]. The air–blood barrier is generated by seeding primary ATI and ATII lung alveolar epithelial cells and primary lung endothelial cells on the upper and lower side of a PDMS- or ECM-like membrane, respectively. Moreover, this device can also recreate physiological breathing movements by applying a tensor force to the membrane, causing mechanical stretching of the alveolar-capillary barrier [31]. The first-generation lung-on-chip systems usually mimic a single alveolus at a time and, therefore, are not able to reproduce the typical alveolar network of the lung tissue. As there is a close relationship between mechanical forces and lung functions, it is essential to generate a network of tiny alveoli that resemble the physiological organ dynamics [32]. However, the second-generation lung-on-chip devices are able to overcome this drawback as they take advantage of a thin gold mesh with an array of hexagonal pores (Alveoli) of about 260 µm in size that serves as a scaffold where an ECM-like membrane of collagen–elastin is polymerized [32]. A more recent chip system has been proposed. It has a miniaturized 96-well-based mid-throughput lung-on-chip system, which allows for the screening of hundreds of biological conditions in a single experiment, opening up a path for the application of this technology in the early phases of drug discovery [33]. Therefore, lung-on-chip technology is a powerful predictive tissue modelling system, which can be used in substitution for animal testing in the preclinical validation phases of innovative lung therapies.

#### 2.2.2. Lung Organoids

Innovative physiologically complex in vitro models for many organ systems have stemmed from three-dimensional (3D) organoid cultures. These systems are capable of studying both organ development and disease. Miller et al. reported on a detailed stepwise differentiation protocol of human pluripotent stem cells (hPSCs) into human lung organoids (HLOs) [34]. HLOs are made up of fetal-like epithelial and mesenchymal compartments organized with structural features that are similar to the native lung, along with populations of cells that express alveolar cell markers. The differentiation process takes a long time, i.e., ≈60 days, and is operator-dependent. Moreover, as is the case with most other 3D organoids, transplantation into a mouse is necessary before full maturation can be achieved. It has been proven that most ATI and ATII cells express SOX9, which is a marker of alveolar progenitor cells, confirming that most alveolar-like epithelium is not yet fully differentiated. This implies that HLOs are not really so good to use for the study of fully differentiated cell function. Even if there are intrinsic limitations, a recent innovation has been optimized and successfully implemented, i.e., in vitro platforms for high-throughput drug screening based on HLOs able to identify novel SARS-CoV-2 inhibitors [35]. Moreover, the same authors characterized the cellular composition of HLOs using single cell RNA-sequencing, further confirming the presence of cells expressing ATI- and ATII-specific markers along with fibroblast and stromal cells [35]. 

Despite the immature phenotype of alveolar epithelial cells, when all this evidence is taken as a whole, lung organoids can provide an alternative to bidimensional primary lung cells in preclinical validation of innovative lung therapies. 

#### 2.2.3. The Use of Synchrotron Imaging in Biomedical Imaging

One of the hardest challenges biomedical imaging has to face is the clarification of the 3D structure and real-time lung function in vivo at a microscopic level. One of the most promising methods, supported by current evidence, is the use of synchrotron radiation-based phase-contrast imaging or K-edge digital subtraction imaging. 

This technique provides dynamic measurements of lung function and allows for the study of regional lung structure [36]. Synchrotron radiation differs from X-ray computed tomography (CT) as it provides not only a quantitative evaluation of the lung morphology but allows for an assessment of the regional lung ventilation, perfusion and inflammation in preclinical animal models of lung diseases [36]. Synchrotron imaging is a unique tool as offers a better understanding of lung pathological re-modelling and can be used to evaluate the efficacy of novel therapeutic strategies proposed to protect lung function under pathological conditions. 

## 3. The Physiological Role of Alveolar Type II Cells

ATII cells have four main functions, i.e., (1) the production and secretion of surfactant, (2) the transepithelial movement of water and ions regulating the volume of the alveolar surface liquid (ASL) preventing alveoli flooding, (3) the expression of immunomodulatory proteins necessary for host defense and the regulation of innate immunity and (4) the regeneration of alveolar epithelium after injury (Table 1).

### 3.1. Production and Secretion of Surfactant

The pulmonary surfactant that lines the alveoli is essential for life as it lowers alveolar surface tension, preventing atelectasis at the end of expiration. ATII cells are unique in their role of synthesizing and assembling all surfactant components (90% lipids and 10% proteins) and storing them in specific organelles (lamellar bodies), and finally, secreting them by exocytosis in the lumen of alveolus. Each ATII cell has about 150 lamellar bodies, with an average diameter of 1 μm and multiple phospholipid bilayers, with a typical unique cellular morphology [13,37,38,39]. 

Several signaling pathways that regulate surfactant secretion have been extensively characterized. These include three distinct signaling mechanisms: (1) the activation of adenylate cyclase which forms cAMP and activates cAMP-dependent protein kinase, (2) the activation of protein kinase C and (3) a Ca^2+^-regulated mechanism that probably leads to the activation of Ca^2+^-calmodulin-dependent protein kinase [13,37,38,39]. These signaling mechanisms are activated by a variety of agonists, such as ATP, which activates all three signaling mechanisms. Although the knowledge as to the identity of several of the signaling proteins involved in surfactant secretion is on constant increase, numerous aspects are still poorly understood. Even if the involvement of the three kinases (protein kinase A, PKA; protein kinase C, PKC; Ca^2+^/calmodulin-dependent protein kinase, CaMK) in the regulation of surfactant secretion has been established, their physiological substrates have not yet been identified. Furthermore, the role and regulation the cytoskeleton plays in lamellar body fusion to the plasma membrane and in the recycling of surfactant material and identification of the molecular mechanisms that couple the rate of recycling to that of secretion, are still a question of debate. These major issues are to be better understood if we are to clarify how secretion and recycling of surfactant are regulated [13,37,38,39].

The surfactant lipids have both hydrophilic and hydrophobic properties (amphipathy) and the head groups have charged qualities that form stable surface-active films at the air–liquid alveolar interface [13,37,38,39]. 

The low surface tension is provided by phospholipids, predominantly dipalmitoylphospha-tidylcholine, and is assisted by the hydrophobic surfactant proteins SP-B and SP-C [37,38,39]. As only ATII cells produce the surfactant protein C, it is considered an ATII cell-specific marker. Moreover, pulmonary surfactant protects the distal lung from noxious particles and microorganisms. 

### 3.2. Transepithelial Movement of Water and Ions Regulating the Volume of the Alveolar Surface Liquid

ATII cells keep the alveolus relatively dry by transporting sodium from the apical surface into the interstitium through the epithelial sodium channel (ENaC) that has three homologous subunits, i.e., the SCNN1, SCNN1B and SCNN1C, and allows for the flow of Na^+^ ions across high-resistance epithelia, maintaining body salt and water homeostasis [40]. ATII cells also express abundant Na/K ATPase (ATP1A1 and ATP1B1) to maintain the intracellular Na and K concentration and have a rich supply of mitochondria which generates ATP. On the other hand, ATI cells have a low mitochondrial density and, even if they also have Na/K transport components, they do not have enough energy to be effective sodium transporters. The active resorption of alveolar fluid is regulated by a chloride channel, named cystic fibrosis transmembrane conductance receptor (CFTR) in ATII cells and chloride intracellular channel protein 5 (CLIC5) in ATI cells [1,2,3,7,8,9]. The process of transepithelial fluid transport and effective alveolar fluid resorption requires an intact epithelium, such as that present in congestive heart failure, differently to what is observed in adult respiratory distress syndrome or severe pneumonia where the epithelium is severely damaged [40].

### 3.3. The Expression of Immunomodulatory Proteins for Host Defense and Regulation of Innate Immunity

While ATI cells are specialized in alveolar gas exchange, ATII cells are the alveolus defenders. External respiration, i.e., the exchange of gases with the external environment, occurs in the alveoli of the lungs and takes place if the diffusion distance between inhaled gas and the red cells is minimized [41,42,43,44]. This can be accomplished by keeping the air passages open (thanks to low surface tension at the air–liquid interface provided by the pulmonary surfactant), the alveolar fluid volume low (through transepithelial sodium transport) and inflammation and oedema absent (through regulation of the innate immune system). As to the innate immune system, non ha senso ATII cells are able to inhibit microbial growth due to their surfactant properties and are also capable of recruiting immune effector cells, including several macrophage populations, as well as secreting a variety of antimicrobial peptides, e.g., b-defensins 2, lipocalin 2 and lysozyme. Moreover, ATII cells neutralize oxidant gases in the alveolar fluid as they provide reducing substances, such as surfactant phospholipids, and reduce glutathione, ascorbate and urate. Furthermore, ATII cells orchestrate pulmonary innate immunity by suppressing or stimulating the macrophage inflammatory response, neutrophils and other immune cells [41,42,43,44,45]. They are also capable of discriminately secreting a variety of chemokines and cytokines, including, IL-1b, IL-1a, IL-6, IL-8, epithelial neutrophil activating peptide-78, growth-related oncogene-α, macrophage inflammatory protein-2, monocyte chemoattractant protein 1, exotoxin and many others. ATII cells also recognize unmethylated bacterial DNA by membrane TLR-9, leading to NF-B activation and the production of IL-6, IL-8 and 2-defensin [41,42,43,44,45].

### 3.4. Regeneration of Alveolar Epithelium after Injury

ATII cells are also considered to be the progenitor cells for the alveolar epithelium. Although lung parenchyma has a low steady-state cell turnover, it has a strong response to injury which triggers the replacement of damaged cells. Self-renewing ATII cells were first recognized as the main adult stem cells responsible for the maintenance of alveolar epithelium homeostasis [7,8,9]. As aforementioned, they function as a progenitor for new ATI and ATII cells during lung repair [45,46].

However, recent lineage tracing studies have identified alternative lineage cell pools that can be recruited for the repair and regeneration of the alveolar epithelium after severe injury. When the occasional death of an ATI cell occurs, the barrier function is maintained by adequate levels of ATII cell proliferation and differentiation into ATI to replace the lost cells. 

Increased mechanical tension forces in the lung are important to promote alveolar regeneration, as shown by the new alveolar regeneration post-pneumonectomy, consequently to the mechanical stress due to the empty controlateral hemithorax [47,48].

The Cdc42/F-actin/MAPK/YAP signaling cascade in alveolar stem cells is essential for the promotion of alveolar regeneration in response to increased mechanical tension in the lung. It has been demonstrated that several signaling pathways are involved in ATI cell proliferation and differentiation, even if the underlying mechanisms that regulate these processes are still a question of debate. ATII cell behavior (quiescence, proliferation and differentiation) is regulated by a combined effect of various signaling pathways: Wnt, Notch, Hippo/Yap, ROS/Nrf2, c-myb, BMP, Cdc42, cytokines (IL-1b, IL-4, IL-13, IL-6), as well as growth factors (TGFβ, FGF, PDGF, EGF, VEGF) produced by different neighboring cells (the epithelium, mesenchymal-derived cells, the airway smooth muscle, neurons and neuroendocrine cells and the endothelium), cell-to-cell contacts, immune cells and the extracellular matrix [19,20,46]. It seems that a sub-lineage of ATII cells expressing Axin2, the transcriptional target of Wnt signaling, plays a key role in repair after an acute alveolar lung injury [19,20,46,49]. Injury-induced IL-1b and HIF1α signaling are essential for the differentiation of ATII cells into ATI cells [49]. A recent single-cell transcriptome analysis of ATII cells after lung injury suggests the importance of the STAT3-BDNF-TRKB axis in modulating alveolar epithelial regeneration [49,50]. Nevertheless, to date, no common factor (e.g., epigenetic) able to synchronize the downregulation of the various pathways involved in ATII differentiation into ATI has yet been identified [19,20,49]. 

In response to loss of ATI and ATII cells following injury, the remaining ATII cells increase their rate of proliferation and differentiation into ATI cells to improve repair. In this setting, cells in a transitional state showing TP53 activation and cellular senescence-like profiles were identified. These cells are characterized by Keratin-8 expression (Krt8) [19,46,49,50,51,52] and have been given various names by different authors: pre-alveolar type-1 transitional cell state (PATS) by Kobayashi [51], alveolar differentiation intermediate (ADI) cells by Strunz [52] and damage-associated transient progenitors (DATP) by Choi [48]. In pulmonary fibrosis, there is an increase of alveolar epithelial cells in the transitional state, accompanied by an increase in myofibroblasts in the lung [50,51]. Cellular senescence appears to be critical in the pathogenesis of several chronic lung diseases, particularly chronic obstructive pulmonary disease (COPD) and idiopathic pulmonary fibrosis (IPF) [50,51,52]. Accumulation of senescent cells leads to cellular dysfunction, failure to repair, impaired immunity and chronic, low-grade lung inflammation, through the SASP response. Regulation of the molecular pathways controlling senescence in lung cells are now being elucidated, including the critical role of miRNAs. This may lead to new therapeutic strategies, such as senolytic therapies that may be given intermittently, inhibitors of PI3K-mTOR signaling, restoration of antiaging molecules and targeting specific miRNAs. Extracellular vesicles may propagate senescence within the lung, but also lead to extrapulmonary spread to account for comorbidities of chronic lung diseases and multimorbidity. The MiRNA in extracellular vesicles (EVs) may also serve as biomarkers for cellular senescence and may be measured in the plasma, sputum and bronchoalveolar lavage (BAL) of patients with chronic lung disease. Novel therapies are likely to have a major effect on the progression of chronic lung diseases and on “health-span” in the future. Therefore, further translational research is strongly encouraged, since delivery of senotherapies by inhalation may be a good way of testing these new approaches [19,50,51,52].

## 4. The Role of ATII Cells in Respiratory Diseases

ATII cells play a crucial role in lung repair/regeneration after injury. Dysfunctional alveolar epithelium is implicated to such an extent in almost every lung disease that, of late, it has been considered a therapeutic target. Therefore, studying this role from a cellular perspective could well provide a novel understanding of lung diseases. ATII dysfunction or dropout has been implicated in the pathogenesis of a variety of parenchymal lung diseases, such as idiopathic pulmonary fibrosis (IPF) and chronic obstructive pulmonary disease (COPD) [13,14,17,18,19] (Table 1).

### 4.1. Idiopathic Pulmonary Fibrosis

As aforementioned, ATII cells are a heterogeneous population that have critical secretory and regenerative roles in the alveolus, maintaining lung homeostasis. However, impairment of their normal functional capacity and the development of a pro-fibrotic phenotype has been demonstrated to contribute to the development of idiopathic pulmonary fibrosis (IPF). A number of factors contribute to ATII cell death and dysfunction. Indeed, during IPF onset, an increased apoptosis of ATII cells was demonstrated both in areas with and without established fibrosis [50,51,52].

ATII cells that escape apoptosis adopt a mesenchymal cell fate by a process referred to as epithelial–mesenchymal transition (EMT) [53]. EMT sustains the generation of pathologically “activated” ATII cells that, by secreting pro-fibrotic factors, further amplifies the fibrotic response, impeding proper alveolar re-epithelialization after injury [53,54,55].

As a mucosal surface, ATII cells are exposed to environmental stresses that can have lasting effects that contribute to fibrogenesis. Genetic risks, that can lead to ATII cell impairment and the development of lung fibrosis, have also been identified. Two general categories of gene mutations were observed in familiar pulmonary fibrosis: genes whose products contribute to specialized secretory functions of ATII cells and those involved in the regulation of stem cell longevity. Aging is a fundamental factor leading to the pathogenic changes observed in ATII cells. 

It has been established that senescent ATII cells are IPF drivers and dysfunctional ATII cells play a pivotal role in lung fibroblast activation and proliferation [15,16,17]. The recent identification of new ATII cell subsets has improved the understanding of how these cells behave in lung development and post-injury repair. The Wnt–β-catenin pathway is known to be activated in the ATII cells of patients with IPF. Diseased ATII cells express the WNT1-inducible-signaling pathway protein 1 (WISP-1), which was shown to induce, on the one hand, ATII proliferation, and, on the other, ATII-EMT and extracellular matrix deposition by lung fibroblasts [49,56,57,58,59,60]. Treatment with WISP1 antibody has been observed to inhibit the development of fibrosis and pharmacological blockade of WISP1 has been reported to exhibit potential as a novel therapy for the inhibition of kidney fibrosis [56,57,58,59,60]. 

These data illustrate three important concepts: firstly, the Wnt–β-catenin pathway influences ATII cell phenotype; secondly, factors secreted by activated ATII cells direct the fibrotic properties of local fibroblasts, and thirdly, the Wnt–β-catenin pathway is an important regulator of pulmonary fibrosis (PF). As most lung fibrosis are of parenchymal nature, it has been postulated that airway epithelial responses are not central mediators of disease [49,56,57,58]. However, emerging evidence suggests that this paradigm may require a revision. Indeed, a proteomics-based study compared the protein profiles of lung biopsies from patients with rapidly progressive IPF to those with stable disease, and it was observed that those with rapidly progressive disease had an enhancement of the mucin family member lPLUNC. The same study also reported that mutations in the MUC5B gene, a mucin family member classically associated with airway epithelial cells, is a risk factor for IPF development. One common single-nucleotide polymorphism (SNP) of MUC5B, the variant rs35705950, is the dominant genetic risk factor for developing IPF [60,61]. MUC5B, a mucin present in conducting airways cells, is also co-expressed with surfactant protein C in ATII cells in healthy and IPF lung tissue. The expression of MUC5B in the epithelial cells lining honeycomb cysts in IPF is associated with prominent bronchiolization of IPF tissue, together with missed alveolar regeneration, a common pathological feature of Usual Interstitial Pneumonia pattern (UIP) [60,61]. 

### 4.2. Chronic Obstructive Pulmonary Disease (COPD)

It has been suggested that ATII cells, physiologically responsible for the production and secretion of surfactant, participate in the development of chronic obstructive pulmonary disease (COPD). This disease is also characterized by an abnormal lung inflammatory reaction. In response to oxidative stress, ATII cells express a number of either anti- or pro-inflammatory genes, including those coding for heamoxygenase-2 (HO-2) and inducible nitric oxidase (iNOS). An upregulation of both these enzymes has been observed in the lungs of patients with severe COPD. How the induction of HO-2 and iNOS contribute to COPD onset has not yet been well understood, as their activity, if not finely tuned, can lead to either protection or apoptosis of ATII cells [62,63]. Although apoptosis is important for the elimination of an excess of activated inflammatory and dead epithelial cells, an excessive apoptosis of airways and alveolar epithelial cells may lead to a reduction in-host defenses [64]. Accordingly, a statistically significant increase in the apoptosis of BAL-derived lymphocytes and brushing-derived airway epithelial cells has been observed in COPD [56,61]. The matrix metalloproteinases (MMPs) and their inhibitors are principal actors in the destructive part of re-modelling events [65,66,67], whereas basic fibroblast growth factor and transforming growth factor-beta (TGF-beta) play a major role in the tissue repair/fibrotic changes. Increased MMP-9, TGF-beta alveolar and ATII cells have been observed in bronchial epithelial cells [68,69]. On the other hand, ATII cells may play a potential protective role in lung repair during chronic disease [70] since they are able to proliferate and transform into ATI and may inhibit fibroblast proliferation and collagen synthesis [69,70,71]. 

Recent studies have reported an increased number of senescent ATII and endothelial cells in COPD patients [64,65,66]. These findings suggest an underlying role of cell senescence in COPD development and, in part, justify the increased prevalence of this disease in older patients. These studies may suggest an exhausted ATII and mesenchymal progenitor cell phenotype in a final common pathway downstream of repeated alveolar repair in the setting of recurring toxic exposure [64,65,66]. Furthermore, a reduced Wnt signaling in the COPD microenvironment has been associated with defective alveolar epithelial repair. It has also been suggested that this defective response is downstream to a shift in canonical to noncanonical Wnt signaling in the presence of toxic stimuli, such as cigarette smoke [67,71,72,73,74]. It still remains a question of debate whether Wnt signaling is associated with an aberrant or absent response by the Wnt-responsive progenitor ATII cell sub-lineage in the ATII cell population [71].

### 4.3. Lung Cancer (Pulmonary Adenocarcinoma)

ATII respiratory epithelial cells are not only essential for normal lung function, but they may also play a crucial role in the pathogenesis of pulmonary adenocarcinoma [71,72,73,74]. Moreover, study data on histology, immunofluorescence and experimental lineage, specific in oncogenic/tumor suppressor gene mouse models, have postulated that ATII cells are the central cells of origin in lung adenocarcinoma and atypical adenomatous hyperplasia (AAH), the only progressive pre-neoplastic change documented in the development of malignant adenocarcinoma [71,72,73,74]. Clinical and experimental observations have recently demonstrated that EGFR-mutant tumors transform into SCLC (small cell lung carcinoma) after EGFR tyrosine kinase inhibitor treatment. This phenomenon suggests that, although these cancers might have common cells of origin, epithelial cell plasticity defines the heterogeneity in lung cancer, taking diverse differentiation cacophonic paths and undergoing different genetic changes during the course of tumor development and progression [71,72,73,74,75].

### 4.4. Acute Respiratory Distress Syndrome 

Acute Respiratory Distress Syndrome (ARDS), a life-threatening lung condition, is associated with diffuse alveolar damage (DAD) and lung capillary endothelial injury. In 1967, Petty et al. used the term “adult respiratory distress syndrome” to describe this condition. The term was later modified, after recognition that this lung condition occurred in patients of all ages, by replacing “adult” with “acute” [76,77,78,79,80,81,82,83]. The early phase is described as being exudative and the later phase fibro-proliferative [76,77,78]. There is an increase in the permeability of the alveolar-capillary barrier in the early phase, which allows for an influx of fluid into the alveoli. As the alveolar-capillary barrier is formed by the microvascular endothelium and the epithelial lining of the alveoli, several conditions that damage either the vascular endothelium or the alveolar epithelium can induce ARDS [76,77,78,79,80,81,82].

The main injury site may be either the vascular endothelium, e.g., sepsis, or the alveolar epithelium, e.g., aspiration of gastric contents. Endothelium injury increases capillary permeability and allows an influx of protein-rich fluid into the alveolar space. Injury to the alveolar lining cells also promotes the formation of pulmonary oedema [77,78,79,80,81,82,83]. 

ATII damage leads to a decreased production of surfactant, interfering with the normal lung repair processes and eventually leading to diffuse lung fibrosis [56,57,58,59].

In summary, there is a need for a better understanding of the physiologic functions of ATI and ATII cells if innovative and effective treatment for ARDS is to be developed.

### 4.5. ATII in SARS-CoV-2 Infection (COVID-19) 

During the SARS-CoV-2 pandemic, the role ATII cells play was evidenced in the most severe COVID-19 cases. Indeed, the COVID-19 patient becomes hypoxic and CT scans evidence the presence of scattered subpleural ground-glass densities in the virus attack phase. ATII cells express both Angiotensin-converting enzyme 2 (ACE2) and the serine protease TMPRSS2 [84,85,86,87,88].

What we do know is that freshly isolated ATII cells may vary in their expression of the ACE2 protein and susceptibility to severe disease [84,85,86]. The infected ATII cells trigger the innate immune response, which favors virus propagation to adjacent alveoli as some ATII cells have apical surfaces in more than a single alveolus. The cell-to-cell transmission from ATII to ATI cells with a rapidly spreading viral replication can make this type of injury explosive, diffusing the damage to the endothelium. A dysregulated inflammatory response leads to a cytokine storm, the hallmark of the most severe SARS-CoV-2-related ARDS [84,85,86,87,88]. This causes fibrinogen and other plasma proteins to leak into the alveoli, where they impair the ability the surfactant has to adsorb to the surface and lower surface tension. Meanwhile, there is an enormous spill-over of pro-inflammatory cytokines into the body. The migration of fibroblast and inflammatory cells into the lumen at the alveolar level causes appositional atelectasis and a loss of gas exchange units. The ATII cell loss means a loss of progenitors for ATI cells, thus impairing ATI regenerative capacity and eventually leading to the progressive worsening of the respiratory function [84,85,86,87,88].

After injury, an exaggerated inflammatory response inside the lung, due to subsequent infection of new areas, may alter the alveolar epithelial cell regulation of fibroblast proliferation and the expression of extracellular matrix genes. Moreover, there may also be an alteration in the normal pathways that usually limit fibrosis through the epithelium [89,90].

## 5. Conclusions

The role ATII cells play in healthy adults and during pulmonary diseases has been extensively elucidated over the last few years, evidencing their central role in homeostasis and acute/chronic illness.

Further studies are ongoing in an effort to contribute to what little is known about these “mysterious cells” and how they go about their seemingly simple everyday tasks.

## Figures and Tables

**Figure 1 ijms-22-02566-f001:**
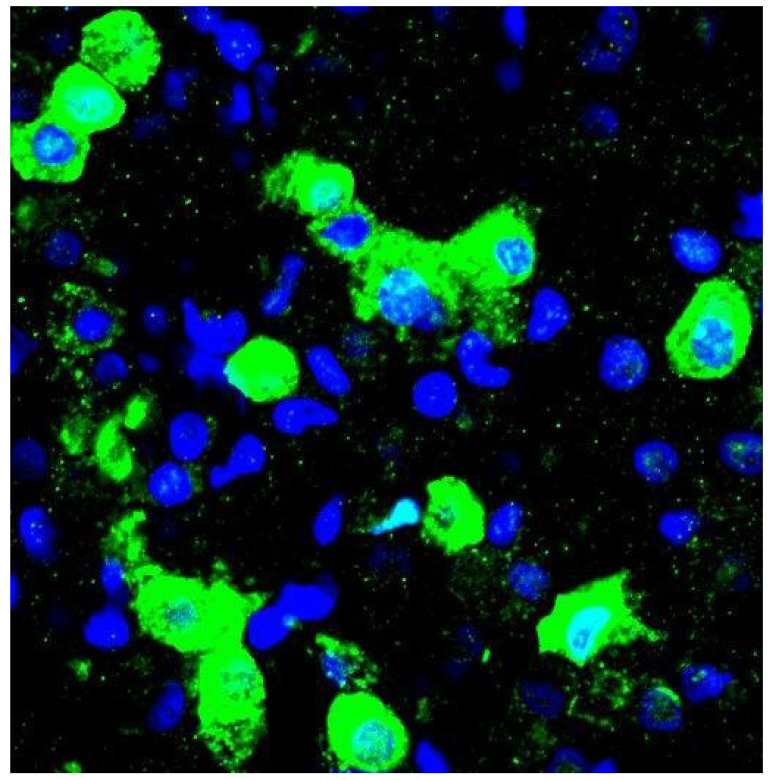
Morphology and protein expression of adult human alveolar type II (ATII) cells. Representative image of immunofluorescence staining for isolated human alveolar type II cells, expanded in culture after 2 days. Alveolar type II cells (ATII) are small and have a cuboidal shape. ATII cells were isolated from five different patients. The cells were seeded after isolation in primary 96-well plates coated with collagen type II. Cell growth in PneumaCult medium enriched with 10% Fetal Bovine serum and 1% Penicillin/Streptomycin. The cells were washed the next day with Phosphate Buffered saline and fixed with Paraformaldehyde at 4% and stained for the specific ATII marker Surfactant Protein C (Pro-SPC) (shown in green). Cell nuclei were stained in blue using Hoechst staining. The image was acquired with a confocal microscope Zeiss LSM 880 with Aryscan. Scale bar 10 μm.

**Figure 2 ijms-22-02566-f002:**
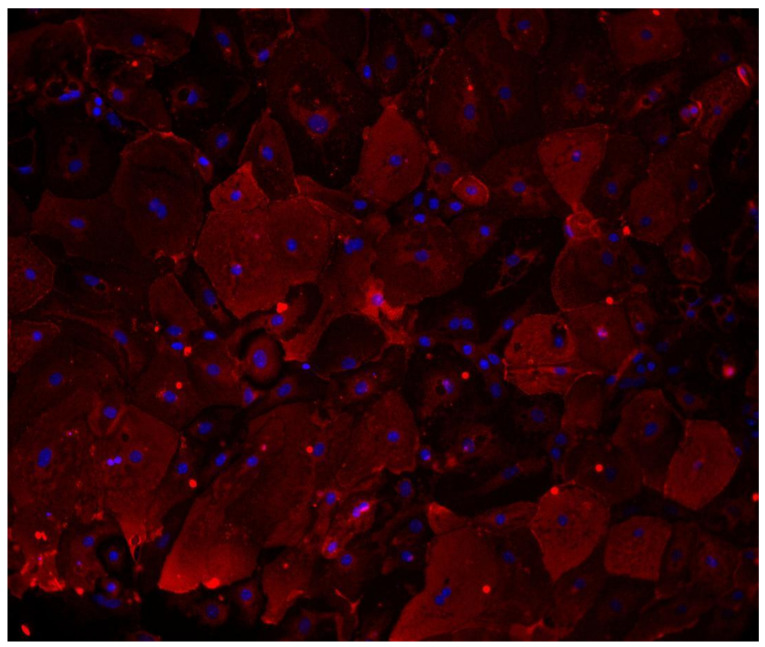
Trans-differentiation of Alveolar type II cells into Alveolar type I (ATI) cells in culture. A representative image of isolated human alveolar type II cells differentiated into alveolar type I cells after 8 days in culture. ATI cells are large and flat cells and occupy 95% of the alveolar compartment. After isolation. The cells were seeded in primary 96-well plates coated with collagen type II in PneumaCult medium, enriched with 1% Pen/Strep. The cells were harvested after 8 days, without FBS, so as to promote trans-differentiation in vitro. The cells were washed with PBS after 8 days in culture and fixed for the specific marker of AT1 cells, Rage (red). There was no evidence of Surfactant Protein C expression (green) after 8 days in culture. The cell nuclei were identified in Blue, using Hoechst staining. The image was acquired by a Nikon Eclipse Ti-E inverted fluorescent microscope, equipped with DC-152Q-C00-FI using NIS V4.30 software (Nikon). Scale bar 100 mm.

**Table 1 ijms-22-02566-t001:** Summary of physiologic and pathologic roles of alveolar epithelial type II (ATII) cells in pulmonary diseases.

Role of ATII Cells	Functions	Summary
Physiological mechanisms	Production and secretion of surfactant	- Pulmonary surfactant reduces alveolar surface tension to prevent atelectasis at the end of expiration - ATII synthesize, assemble, store and finally secrete surfactant
	Transepithelial movement of water and ions regulating the volume of the alveolar surface liquid	- ATII transports sodium from the apical surface into the interstitium through the epithelial sodium channel (ENaC) and allows for the flow of Na^+^ ions across high resistance epithelia - ATII cells also express abundant Na/K ATPase to maintain the intracellular Na and K concentration and have a rich supply of mitochondria which generates ATP
	Expression of immunomodulatory proteins for host defense and regulation of innate immunity	- ATII cells are the alveolus defenders - ATII cells are capable to inhibit microbial growth, recruit immune effector cells and secrete a variety of antimicrobial peptides - ATII cells neutralize oxidant gases in the alveolar fluid
	Regeneration of alveolar epithelium after injury	- ATII cells are also considered to be the progenitor cells for the alveolar epithelium - In response to loss of ATI and ATII cells following injury, the remaining ATII cells increase their rate of proliferation and differentiation into ATI cells
Respiratory diseases	Idiopathic pulmonary fibrosis (IPF)	- During IPF, an increased apoptosis of ATII cells was demonstrated both in areas with and without established fibrosis - ATII cells that escape apoptosis become pathologically “activated” ATII cells that, by secreting pro-fibrotic factors, further amplifies the fibrotic response, thus impeding proper alveolar re-epithelialization
	Chronic obstructive pulmonary disease (COPD)	- COPD is characterized by an abnormal lung inflammatory reaction - ATII cells express a number of either anti-or pro-inflammatory genes, including those coding for heamoxygenase-2 (HO-2) and inducible nitric oxidase (iNOS), that are upregulated in the lungs of patients with severe COPD - The number of senescent ATII are increased in COPD patients
	Lung cancer (pulmonary adenocarcinoma)	- ATII cells are the central cells of origin in lung adenocarcinoma and atypical adenomatous hyperplasia (AAH), the only progressive pre-neoplastic change documented in the development of malignant adenocarcinoma
	Acute respiratory distress syndrome	- ATII damage leads to a decreased production of surfactant, interfering with the normal lung repair processes and eventually leading to diffuse lung fibrosis
	SARS-CoV-2 infection (COVID-19)	- Transmission of virus from infected ATII cells to ATI cells with a rapidly spreading viral replication can make this type of injury explosive, diffusing the damage - ATII cell loss means a loss of progenitors for ATI cells, thus impairing ATI regenerative capacity and eventually leading to the progressive worsening of the respiratory function

## Data Availability

No new data were created or analyzed in this study. Data sharing is not applicable to this article.

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
