# Peer review of "The History and Mystery of Alveolar Epithelial Type II Cells: Focus on Their Physiologic and Pathologic Role in Lung"

_ijms, 2021, doi:10.3390/ijms22052566_

Round 1

Reviewer 1 Report

This review focused on the physiologic and pathologic roles of type II alveolar epithelial cells (ATII). The review was very well written and contained the details of the physiologic functions of type II alveolar epithelial cells and how these cells exerted repair/regeneration in a few common lung diseases.

Just one comment, the authors discussed about the scientific advances in lung anatomy in section 2, however, new technologies to explore the function and underline mechanism of ATIIs are worthwhile to be included. For example, Lung-on-a-chip, 3D organoid culture and imaging of lung function using synchrotron. These are important to investigate the physiological and pathologic roles of ATIIs.

Author Response

R: We would like to thank the reviewer for these comments that have enabled us to enhance our manuscript . We have added some comments as to this topic in the 2.2  section (lines 146-202):

2.2 New technologies to explore the function of alveolar epithelial cells

Lung on Chip

Organs-on-Chips are micro-engineered cell-based systems developed to resemble the multicellular dynamics, cell-cell interactions, vascular perfusion and microenvironments of real organs. The lung on chip is a biomimetic micro-system able to reconstruct the functional alveolar epithelial-capillary interface (31). The air-blood barrier is generated by seeding primary ATI and ATII lung alveolar epithelial cells and primary lung endothelial cells on the upper and lower side of a PDMS or ECM-like membrane, respectively. Moreover, this device can also recreate physiological breathing movements by applying a tensor force to the membrane, causing mechanical stretching of the alveolar-capillary barrier (31). The first generation lung-on-a-chip systems usually mimic a single alveolus at a time and, therefore , are not able to  reproduce the typical alveolar network of the lung tissue. As there is a close relationship between mechanical forces and lung functions, it is essential to generate a network of tiny alveoli that resemble the physiological organ dynamics (32). However, the second generation  lung-on-Chip devices are able to overcome this drawback as they take advantage of a thin gold mesh with an array of hexagonal pores (Alveoli) of about 260 µm in size that serves as a scaffold where an ECM-like membrane of collagen–elastin is polymerized (32). A more recent chip system has been proposed. It has a miniaturized 96-well-based mid-throughput lung on a chip system, which allows for the screening of hundreds of biological conditions in a single experiment, opening up a path for the application of this technology in the early phases of drug discovery (33). Therefore, lung-on-a-chip technology is a powerful predictive tissue modelling system, which can be used in substitution for animal testing in the preclinical validation phases of innovative lung therapies.

Lung Organoids

Innovative physiologically complex in vitro models for many organ systems have stemmed from 3D organoid cultures. These systems a capable of studying both organ development and disease. Miller et Al. reported on a detailed step-wise differentiation protocol of human pluripotent stem cells (hPSCs) into human lung organoids (HLOs) (34). HLOs are made up of fetal-like epithelial and mesenchymal compartments organized with structural features that are similar to the native lung, along with populations of cells that express alveolar cell markers. The differentiation process takes a long time , i.e. ≈60days and operator dependent: Moreover, as is the case with most other 3D organoids, transplantation into a mouse is necessary before full maturation can be achieved. It has been proven that most ATI and ATII cells express SOX9, which is a marker of alveolar progenitor cells confirming that most alveolar-like epithelium is not yet fully differentiated. This implies that HLOs are not really so good to use for the study of fully differentiated cell function. Even if there are intrinsic limitations,  a recent innovation has been optimized and successfully implemented i.e. in vitro platforms for high-throughput drug screening based on HLOs able to identify novel SARS-CoV-2 inhibitors (35). Moreover, the same authors characterized the cellular composition of HLOs using single cell RNA-sequencing, further confirming the presence of cells expressing ATI and ATII specific markers along with fibroblast and stromal cells (35). Despite the immature phenotype of alveolar epithelial cells, when all this evidence is taken as a whole, lung organoids can provide an alternative to bidimensional primary lung cells in preclinical validation of innovative lung therapies.

The use of synchrotron imaging in biomedical imaging

One of the hardest challenges biomedical imaging has to face is the clarification of the 3D structure and real-time lung function in vivo at a microscopic level. One of the most promising methods, supported by current evidence, is the use of synchrotron radiation-based phase-contrast imaging or K-edge digital subtraction imaging.

This technique provides dynamic measurements of lung function and allows for the study of regional lung structure (36). Synchrotron radiation differs from X-ray computed tomography (CT) as it provides not only a quantitative evaluation of the lung morphology but allows for an assessment of the regional lung ventilation, perfusion and inflammation in preclinical animal models of lung diseases (36). Synchrotron imaging is a unique tool as offers a better understanding of lung pathological remodelling and can be used to evaluate the efficacy of novel therapeutic strategies proposed to protect lung function under pathological conditions.

Reviewer 2 Report

In the manuscript titled “The history and mystery of alveolar epithelial type II cells: focus on their physiologic and pathologic role in lung,” Ruaro and colleagues summarized the physiologic and pathologic roles of alveolar type II (ATII) cells in pulmonary diseases. They also included the key signaling pathways involved in cell growth, differentiation, inflammatory response, fibrosis, etc. The content covers the current knowledge of ATII and the references are up to date. The manuscript is well-organized and written clearly to the audience. But some minor points are observed:

  1. Figure legends are concise. More details should be included, such as cultural conditions. Scale bar is also needed.
  2. Line 232, “AT1” should be “ATI”.
  3. In section 3.1 Production and secretion of surfactant, it would be more helpful if more details in lipids are discussed. The signaling pathways that regulate the secretion of surfactant were not mentioned.
  4. I would suggest the authors add a table showing the key physiologic and pathologic roles of ATII cells in pulmonary diseases discussed in the review.

Author Response

  1. Figure legends are concise. More details should be included, such as cultural conditions. Scale bar is also needed.

R: We would like to thank the reviewer  this comment.  As requested, more details have been included in the Figure legends (Figure 1 lines 88-95 and Figure 2 lines 97-105):

Figure 1. Morphology and protein expression of adult human alveolar type II cells. Representative image of immunofluorescence staining for isolated human alveolar type II cells, expanded in culture after 2 days. Alveolar type II cells (ATII) are small and have a cuboidal shape. ATII cells were isolated from five different  patients. The cells were seeded after isolation in primary 96 well plates coated with collagen type II. Cell growth in PNEUMA-CULT medium enriched with 10% FBS and 1% Pen/Strep. The cells were washed the next day with PBS and fixed with PFA at 4% and stained for the specific ATII marker Surfactant Protein C (Pro-SPC) (shown in green). Cell nuclei were stained in blue using Hoechst staining. The image was acquired with a confocal microscope Zeiss LSM 880 with Aryscan. Scale bar 10 mm. 

Figure 2. Trans-differentiation of Alveolar type II cells into Alveolar type I cells in culture.

A representative image of isolated human alveolar type II cells differentiated into alveolar type I cells after 8 days in culture. ATI cells are large and flat cells and occupy 95% of the alveolar compartment. After isolation. The cells were seeded in primary 96 well plates coated with collagen type II in PNEUMA-CULT medium, enriched with 1% Pen/Strep. The cells were harvested after 8 days, without FBS, so as  to promote trans-differentiation in vitro. The cells were washed with PBS after 8 days in culture and fixed for the specific marker of AT1 cells Rage (red). There was no evidence of Surfactant Protein C expression (green) after 8 days in culture. The cell nuclei were identified in Blue, using Hoechst staining. The image was acquired by a Nikon Eclipse Ti-E inverted fluorescent microscope, equipped with DC-152Q-C00-FI using NIS V4.30 software (Nikon). Scale bar 100 mm. 

  1. Line 232, “AT1” should be “ATI”.

R: Thank you for pointing out this typo, it has been corrected.

  1. In section 3.1 Production and secretion of surfactant, it would be more helpful if more details in lipids are discussed. The signaling pathways that regulate the secretion of surfactant were not mentioned.

R: We have added further details as requested  (lines 291-236):

Several signaling pathways that regulate surfactant secretion have been extensively characterized. These include  three distinct signaling mechanisms: 1: the activation of adenylate cyclase which forms cAMP and activates cAMP-dependent protein kinase; 2: the activation of protein kinase C; 3: a Ca2+ - regulated mechanism that  probably leads to the activation of  Ca2+-calmodulin-dependent protein kinase (13,32). These signaling mechanisms are activated by a variety of agonists, such as ATP which activates all three signaling mechanisms. Although the knowledge as to the identity of several of the signaling proteins involved in surfactant secretion is on constant increase, numerous aspects are still poorly understood. Even if the involvement of the three kinases (protein kinase A, PKA; protein kinase C, PKC; Ca2+/calmodulin-dependent protein kinase, CaMK) in the regulation of surfactant secretion has been established, their physiological substrates have not yet been identified. Furthermore, the role and regulation the cytoskeleton plays in lamellar body fusion to the plasma membrane and in the recycling of surfactant material and identification of the molecular mechanisms that couple the rate of recycling to that of secretion, are still a question of debate. These major issues are to be better understood if we are to clarify how secretion and recycling of surfactant are regulated (13,37,38)”.

  1. I would suggest the authors add a table showing the key physiologic and pathologic roles of ATII cells in pulmonary diseases discussed in the review.

R: We thank the reviewer for this comment , which allows us to enhance the quality of the manuscript.  Table 1 has been added to this aim.

Table 1. Summary of physiologic and pathologic roles  alveolar epithelial type II (ATII) cells play in pulmonary diseases

Role of ATII cells

Functions

Summary

Physiological mechanisms

Production and secretion of surfactant

- Pulmonary surfactant reduces alveolar surface tension to prevent atelectasis at the end of expiration

- ATII cells synthesize, assemble,  store and finally secrete surfactant

Transepithelial movement of water and ions regulating the volume of the alveolar surface liquid

- ATII cells transport sodium from the apical surface into the interstitium through the epithelial sodium channel (ENaC) and allow for the flow of Na+ ions across high resistance epithelia

- ATII cells also express abundant Na/K ATPase to maintain the intracellular Na and K concentration and have a rich supply of mitochondria, which generates ATP

Expression of immunomodulatory proteins for host defense and regulation of innate immunity

- ATII cells are the alveolus defenders

- ATII cells are capable of inhibiting microbial growth, recruiting immune effector cells and secreting a variety of antimicrobial peptides

- ATII cells neutralize oxidant gases in the alveolar fluid

Regeneration of alveolar epithelium after injury

- ATII cells are also considered to be the progenitor cells for the alveolar epithelium

- In response to loss of ATI and ATII cells following injury, the remaining ATII cells increase their rate of proliferation and differentiation into ATI cells

Respiratory diseases

Idiopathic pulmonary fibrosis (IPF)

- During IPF, an increased apoptosis of ATII cells was demonstrated both in areas with and without established fibrosis

- ATII cells that escape apoptosis become pathologically “activated” ATII cells that, by secreting pro-fibrotic factors, further amplify the fibrotic response, impeding proper alveolar re-epithelialization

Chronic obstructive pulmonary disease (COPD)

- COPD is characterized by an abnormal lung inflammatory reaction

- ATII cells express a number of either anti- or pro-inflammatory genes, including those coding for heamoxygenase-2 (HO-2) and inducible nitric oxidase (iNOS), that are upregulated in the lungs of patients with severe COPD.

-COPD patients have an increase in the number of ATII cells

Lung cancer (pulmonary adenocarcinoma)

- ATII cells are the central cells of origin in lung adenocarcinoma and atypical adenomatous hyperplasia (AAH), the only progressive pre- neoplastic change documented in the development of malignant adenocarcinoma

Acute respiratory distress syndrome

- ATII damage leads to a decreased production of surfactant, interfering with the normal lung repair processes and eventually leading to diffuse lung fibrosis

SARS-CoV-2 infection (COVID-19)

- The transmission of viruses from infected ATII cells to ATI cells with a rapidly spreading viral replication can make this type of injury explosive, diffusing the damage

- ATII cell loss means a loss of progenitors for ATI cells, impairing the ATI regenerative capacity and eventually leading to the progressive worsening of the respiratory function
